# Building Resilience: The Critical Role of Multicultural Education to Cope with the Psychological Trauma of Migrant and Refugee Youth

**DOI:** 10.3390/bs15070916

**Published:** 2025-07-07

**Authors:** Lito Nantsou, Zoe Karanikola, Nektaria Palaiologou

**Affiliations:** Language Education for Refugees and Migrants (L.R.M.) Programme, Department of Education, Hellenic Open University, Perivola Patron, 26335 Patras, Greece or zoekar@upatras.gr (Z.K.); nekpalaiologou@eap.gr (N.P.)

**Keywords:** multicultural education, migrants, refugees, inclusion, trust, psychological trauma

## Abstract

Enhancing and developing multicultural education is essential in today’s interconnected world, characterized by significant migration and refugee movements, as it serves as a fundamental element for effectively integrating migrant and refugee students within host nations. In Greece, which has long dealt with the challenge of integrating thousands of individuals fleeing conflict and disasters, there is a pressing need to evolve and modernize this educational approach. This qualitative study, based on semi-structured interviews with nine multicultural educators in Greece, seeks to explore how teachers foster a sense of belonging and inclusion in their diverse classrooms. Despite facing systemic challenges, the findings reveal that educators strive to highlight students’ cultural heritages through collaborative group activities that encourage trust, respect, and appreciation for diversity. Additionally, the research delves into how teachers cope with the psychological trauma often experienced by these students. Participants expressed that the inconsistent availability of specialists and monitoring for students dealing with severe psychological issues complicates their teaching efforts.

## 1. Introduction

The European Union is currently confronting a significant challenge, known as the Anthropocene crisis, and is called to seek solutions for strongly interconnected issues, such as socio-economic crises, climate change, the COVID-19 pandemic, and social disparities, which threaten individuals’ well-being and jeopardize the concept of democracy and human rights ([15]). In addition, in the early 21st century, there has been a significant increase in migration and refugee numbers, with individuals fleeing conflict, persecution, economic struggles, and environmental disasters seeking asylum in Western countries. Specifically, Europe, in the aftermath of the 2015 humanitarian crisis, experienced an unprecedented number of people seeking asylum, which eventually led to the EU-Turkey agreement in 2016 ([16]), the closure of the Balkan route, and disproportionate migratory pressure on first asylum states, including Greece, the focus of this article. These pressures increased after the onset of renewed military operations involving Russia and Ukraine in February 2022 ([28]). At the national level, migration inflows in recent years have also been impressive, with the number of refugees exceeding 160,000, while in 2015 alone, some 860,000 refugees arrived in Greece. The total number of refugees in Greece in 2024 was 62,119 ([23]). Greece, as a main entry point to Europe, has faced challenges in integrating these populations, resulting in discontent among local residents dealing with a prolonged economic crisis. This situation creates pressure on institutions and can foster xenophobia. To ensure effective integration and protect the rights of migrants, particularly refugees, it is essential to uphold international law ([32]).

In this vein, this qualitative research study seeks to investigate the ways teachers attempt to build a sense of belonging and inclusion in their multicultural classes, and at the same time, it aims to explore the methods that educators apply to deal with the possible psychological trauma that could emerge from pupils with refugee and marginalization backgrounds.

Education in the contemporary multicultural context is crucial for facilitating the inclusion of refugees and helping them become active citizens, though various systemic, linguistic, and cultural barriers complicate this process. In addition, the concept of multiculturalism focuses on the integration, respect, and recognition of cultural differences, while the core principles are to connect with the world, bridge differences, support individual social mobility, recognize different cultures, and attribute a positive connotation to diversity ([34]). In this sense, multicultural classrooms adopt inclusive teaching practices, encourage students to speak their home language and bring elements of their culture, facilitate the inclusion of all students, improve their learning outcomes, bridge the gap in teacher–student and teacher–parent relationships, mitigate cultural misunderstandings, and prevent the outbreak of racist behaviors ([25]).

To this end, the Greek Ministry of Education has adopted and implemented various measures aimed at supporting the educational and social integration of all students at both the primary and secondary school levels. More specifically, the official version of Intercultural Education is expressed in Law 2413/96 ([24]), which concerns its purpose and content, Intercultural Education schools, teaching staff, and school administration. In addition, Law 3879/2010 established the Priority Education Zones (ZEPs), which function in areas with attributes such as low overall educational index, high school drop-out rate, low access to higher education, and low socio-economic indicators. These classes are aimed at students with an immigrant and/or refugee background, offered in morning mainstream schools, with a parallel mix of intensive Greek language courses and attendance in regular classes. Furthermore, reception and education structures for refugees were established, which constituted separate classes exclusively for newly arrived refugees, accommodated in school premises ([40]).

However, the full range of possibilities and options provided by the legislative framework does not seem to be sufficiently utilized. Thus, while the legislation provides for the provision of the possibility of teaching the mother tongue and the culture of the country of origin as well as the possibility of hiring specialists (sociologists, psychologists, social workers) to support the work of the Reception and Education Classes, in practice these regulations are applied selectively and on a very limited scale ([33]). Consequently, the monocultural and monolingual perspective continues to be dominant in the teaching of students in both primary and secondary education, whereas the cultural and linguistic capital brought by students from different backgrounds is often ignored ([42]).

In addition, teachers often encounter many obstacles in trying to support the academic and psychosocial development of all students through innovative and inclusive approaches, viewing multicultural education as an opportunity to enrich the learning experience for both teachers and students, mainly due to the lack of initial education and training they have received ([14]). Greek teachers require targeted training to develop a greater respect for diversity ([45]; [46]). The determination and resilience shown by students in overcoming challenges have, in many cases, challenged the prejudices of teachers towards students deriving from different cultural backgrounds and motivated them to enhance their teaching methods. To effectively meet the diverse needs and challenges of a multicultural classroom, it is essential for teachers to engage in ongoing professional development and adopt modern teaching strategies and digital tools ([38]; [30]; [29]; [31]).

In light of this, Geneva Gay’s extensive research emphasizes the significance of integrating students’ cultural backgrounds into the learning process, as this approach highlights cultural identities and fosters academic success, diversity, and equity ([20]). In this context, one of the most impactful strategies for promoting the academic and psychosocial growth of foreign-speaking and often marginalized students is Jim Cummins’ identity texts. These are innovative, multimodal, and collaborative works produced by students, inspired by their experiences and cultural backgrounds. Written in both their home language and the language of the host country, these dual-language books empower students and validate their identities ([12]; [10]). Additionally, acknowledging and leveraging students’ existing linguistic skills enhances their language acquisition ([11]) and assists teachers in countering cultural deficit thinking. Deficit thinking is often considered to be a major obstacle to reducing long-lasting inequalities in Western education, and concerns overgeneralizations about students’ family backgrounds and the adoption of dominant monocultural views about the criteria for school success for students who do not belong to the dominant cultural group ([36]).

In addition, multicultural educators frequently encounter the psychological distress that refugee students experience. To help these students feel as secure as possible and support their cognitive and psychosocial development, it is essential to address this distress from an educational standpoint ([39]). A comprehensive pedagogical approach involves offering multi-faceted empowerment and support for traumatized and marginalized students ([40]). [52] ([52]) investigates trauma and post-traumatic stress disorder, highlighting how they can disrupt both physical and psychosocial functions and impair the brain’s normal operations by affecting perception, memory, and emotional stability.

Abiding by that, [26] ([26]) certify with their research that post-traumatic stress slows down cognitive development, as it causes psychological instability, lack of concentration, reduced self-control, and ability to concentrate, and they emphasize the need to implement culturally sensitive assessments to establish a climate of respect and safety that will favor the development of students. Thus, school can be a means of strengthening mental resilience and adaptability.

Finally, [48] ([48]) focus on educational interventions aimed at empowering student war survivors. They distinguish them into three categories depending on their beneficial effects on the way students think and express their emotions, as well as on the multimodal effort to meet their needs. Although the proposed measures are not always effective, as traumatic experiences may be multiple and hidden, targeted interventions are likely to improve the psychosocial and cognitive state of victims.

## 2. Materials and Methods

This qualitative research study seeks to investigate the ways teachers attempt to build a sense of belonging and inclusion in their multicultural classes, and at the same time, it aims to explore the methods that educators apply to deal with the possible psychological trauma that could emerge from pupils with refugee and marginalized backgrounds.

### 2.1. Data Collection

Primary data collection was based on semi-structured interviews with nine multicultural education teachers, who were invited to openly share their subjective experiences and express their concerns and feelings. It is a method that favors the emergence and detection of inner thoughts, behaviors, and trends that would be more difficult to identify with other qualitative methods. As the questions do not provide possible answers desired by the researchers, they leave the field of expression of the interviewees free ([3]; [37]). Digital applications, such as Skype and Turboscribe, were used to record and transcribe the interviews. The participants were aware that they were being recorded, a fact that is also indicated by the digital means that record the start and end of the recording. The aim was to uncover the strategies they use to support their students’ cultural identities in the classroom and the ways of managing the trauma that may emerge.

The interview protocol consisted of questions divided into four main axes, with a total number of ten open and closed-ended questions. The first two were socio-demographic questions concerning the participants’ profile (age, studies, years of experience in multicultural classrooms) and their students’ profile as well (nationality, immigrant/refugee background). The second group contained three questions regarding the social and emotional needs young immigrant and refugee students have, the importance of their fulfillment, as well as the methods teachers use to identify them. In the third group, there were three questions about approaches teachers embrace and activities they implement to establish an inclusive environment for their students. Finally, the fourth group included two questions regarding trauma management and the main obstacles to its effective treatment.

### 2.2. Data Analysis

The collected data were analyzed thematically to identify and record patterns emerging from the participant interviews. Thematic analysis is a flexible qualitative method ([6]) that, although it involves the risk of subjective interpretation of the data, allows the emergence of common ideas and concerns of the interviewees ([50]). However, the approach is deductive as the interviews are based on predetermined topics that the researcher seeks to discover. The thematic analysis involves the following six steps: familiarizing with the interview data through their in-depth study so the researcher can understand the emerging themes; identifying the interviewees’ common phrases or sentences to decode the data (these elements constitute codes that demonstrate the dominant issues they experienced); highlighting from the extracted codes the themes that are answers to the given questions; checking the correspondence between the data and the themes that arose to ensure consistency and coherence in the conclusions drawn; accurately defining and naming the emerging themes and their relation to the research topic; writing the analysis by synthesizing the common themes and drawing conclusions from them ([6]).

### 2.3. Participants

Nine Greek teachers (N = 9) who have sufficient experience in multicultural education were selected as an accessible and appropriate population through a convenience sampling technique to participate in this investigation ([37]). Due to their relative working experience, they have been called upon to manage the potential emerging trauma from their students with refugee backgrounds. All the participants were female, eight out of nine worked in public primary schools in Greece, one of them in a refugee camp, whereas their age ranged from 28 to 57 years old. It is worth noting that six out of nine participants have work experience teaching in multicultural classrooms and have attended intercultural seminars. Considering that the participants mostly worked in primary education, the ages of the pupils they work with are 6–12 years old.

### 2.4. Ethical Considerations

Critical ethical issues were considered as well. First, the researchers relied on the educators’ voluntary participation, assuring them they could withdraw from the research if they wished. They were also informed of the purpose of the study and assured of anonymity and confidentiality regarding their data. Their identities were ensured, and their personal experiences were not made public. In this way, freedom of expression and a willingness to share their experiences from the multicultural class were sought. The aforementioned practices encourage trust between the researchers and the participating educators. Confidence is crucial so the interviewees can honestly share their reality without self-censorship ([3]).

## 3. Results

Various findings emerged from the interviews, organized according to the research objectives and questions. The main axes of the research are the following two: (a) multicultural education and inclusion, and (b) trauma management and obstacles.

### 3.1. Multicultural Education and Inclusion

One of the most important goals of education is to cultivate smooth socialization and inclusion of students. According to the research, inclusion can be developed through cultural exchanges and shared activities, by building a climate of trust, and by encouraging multilingualism and family engagement. Specifically:

The analysis of the responses demonstrates the positive attitude of teachers towards the multiculturalism that prevails in their classes, since they highlighted diversity to develop inclusion (P1, P8). Elements of the culture of their migrant and refugee students, such as language, history, habits, celebrations, games, and the foods of their people, are transformed into means of communication (P2, P4, P9). At the same time, they sharpen students’ interest in actively participating in the learning process and enrich them with diversity. In particular:

P1: “I mainly tried from the beginning to show that everyone comes from somewhere. They have a story. They speak a language at home.”

P2: “…I think that by demonstrating their identity, linguistic and cultural, the process is definitely strengthened.”

However, participants 3 and 4 pointed out the difficulty of cultivating teamwork and inclusion in multicultural classes due to the students’ multiple differences, culture, and class.

The participants highlighted the value of group activities, especially the playful nature of teaching, for students to overcome the obstacles that cultural diversity may cause. In this context, the role of projects related to vital needs, such as food, was also emphasized, as well as Identity Texts that highlight aspects of the student’s identity (P5, P8). Barriers such as different languages and cultures are removed through the experience that play and art offer (P2, P8). Finally, P2 highlighted the need to create a routine that all students in the class will apply, which increases the students’ sense of security. In particular:

P5: “They can work together only through teamwork, that is, we work with groups. From there and then other techniques that I have seen other teachers do in the field are organizing some events with food from the places where each student is. For example, this event could have a theme of bread. So, everyone will bring either bread or puff pastry from their place.”

P8: “I also use identity texts; that is, I let the children write or draw or present something about themselves. So, we need to get to know each other. Then, the team spirit comes from what they mentioned above.”

P9: “Holidays, let’s say, are a great opportunity to ask about the customs and traditions of other countries, even if it concerns another religion. Let’s say what happens here or if the children celebrate a specific period of time, a religious holiday of their own, to explain how it happens, and why it happens too. Regarding the diversity of the other nationalities. This is mainly what I do with holidays, customs, and traditions. Games, we talk a lot about games, if they are common. What else can I think of? Maybe through some videos, it takes time for some of the children, but not all. It takes time for them to integrate and open up, to say even these simple things.”

Group activities, however, proved to be difficult in some cases. This is because they may bring to the surface underlying tensions and conflicts between students from different cultural backgrounds. Furthermore, these are time-consuming procedures that the school curriculum does not provide for (P4, P5).

The educators emphasized the importance of building trust between class members. Strategies, such as creating a “class contract” (both students and teachers co-decide and commit to how they will behave and function in the classroom), foster a sense of security and mutual trust. At the same time, they promote mutual assistance to solve problems that arise, as well as familiarization with the diversity that governs the multicultural class.

P1: “We must integrate the children and from there the children themselves create a situation in which they will include the children who do not speak in the groups, in their groups with the technique of assistance.”

P8: “Then, the class contract helps a lot, because the children can work in groups, discuss, sit in groups first, and think about how exactly they should function as a group. The class contract creates for them the concept of a group.”

### 3.2. Multilingualism and Inclusion

The teachers involved in the study reported that they apply translanguaging and allow students to use their native language during the learning process. Although some initially questioned whether this practice was effective for learning the target language, the majority of educators support using the students’ first language as a way to approach new knowledge. Based on their experience, some participants have discovered that students’ linguistic abilities serve as a valuable tool for understanding the language and culture of the host country, and for helping them integrate into it. Many emphasized that the sense of acceptance students feel serves as a motivation for their cognitive and social growth (P2, P5, P6). Additionally, they noted that students acting as interpreters for classmates with less proficiency in the host language promotes collaboration among students and makes the teacher’s work easier (P2, P8).

P1: “Of course, and let’s say now I have two little siblings who speak Arabic. The girl started to understand much faster than the boy, of course, I asked her to serve as an interpreter for her brother. Of course, I also used Google Translate…”

P8: “…when we learn new words, I will ask children to translate them in their mother tongue…So, they feel included, and now a multilingual environment is created. And it is very helpful for all…”

The multilingual teaching model, however, presents significant challenges. At the start of their careers, some teachers were initially skeptical and followed the prevailing monolingual teaching model in Greece. However, their experiences in multicultural school environments and their training in intercultural education led them to reconsider their earlier views.

On the other hand, finding the right balance between using students’ linguistic skills and the target language was cited as a difficult goal to achieve. For convenience, students often prefer to use their native language (P1). Coupled with the curriculum set by the Ministry of Education and its rigid teaching schedule, this increases the pressure on teachers, which may slow down the acquisition of the target language. Additionally, when students with similar language backgrounds act as interpreters, there’s a risk that knowledge transmission may be incomplete, a situation that teachers may struggle to detect (P3). Particularly:

P1: “Within the context of the classroom, if I translate all I say in Greek in one way into Arabic, I do not consider it particularly useful. Because the child expects to have help in his/her language and does not make an effort to acquire it. That is, the times I tried it, I saw that it did not help much.”

P3: “When you have a student who doesn’t speak Greek, it’s more important to help him/her learn the alphabet, learn to write, learn the basic vocabulary than to maintain their identity. This will happen in the second year when a basic code of communication has already been mastered… the school time is very limited anyway, … Easter, Christmas holidays, Halloween, holidays anyway, the time is very limited…”

Some of the participants attempted to involve the students’ families in the learning process. This happens mainly when students face particular difficulties and their guardians are called by the school, or the latter wish to assist in the teacher’s effort to help their child. This practice may strengthen ties with teachers and schools and establish a climate of mutual trust.

P1: “…and a very close relationship was developed with the student, as well as with his mother who wanted to have a very personal relationship, to thank us as a school for listening to her child and sharing, in any case, the whole heavy psychology that the child had, still remembering everything he went through.”

P2: “…I believe that it has a great benefit for parents, because interpreting is something very difficult to do … it also brings them face to face with the learning process …”

The participation, however, of the family members proved to be a great challenge. The participating educators pointed out the dysfunctionality of these families due to their migrant and refugee backgrounds (P3, P4). Their painful stories, which often include the loss of parents and guardians, prevent teachers from trying to involve the family in the educational reality. The suspicion of families towards the educational system of the host country, which constitutes an unknown factor in shaping their children, was also highlighted. Finally, the pressing time frame set by the Ministry of Education does not favor similar initiatives that require time and willingness to communicate on the part of teachers (P8).

P4: “…I have noticed that because they are usually painful stories, they do not want to communicate them. In other words, they are closed in this regard. They say very little about the past, very little about relatives, with exceptions of course, but most of them say nothing, mention nothing. So, I do not involve them.”

P8: “Nevertheless, there are basic difficulties. First of all, it requires some preparation and a lot of work for the parent… to be able to enter the school community. Many times, they feel a… they have a reservation towards it. Okay, despite the fact that they respect the teacher…but to enter the school, if the school does not take the first step, it is difficult. And the school, on the other hand, doesn’t always open these doors, because it’s very closed, according to my personal opinion. There’s so much workload at school, such a pressing schedule, that you don’t have the opportunity… Yes, I mean, you have to get a subject out, for example… You have to go on trips, have meetings with parents. And at the same time, you run to maybe two or three schools.”

Most of the participants pointed out that they attempt to foster a climate of trust and inclusion by demonstrating particular empathy in the management of trauma that may arise. They emphasized the importance of discretion in the management of students’ sensitive personal data, prioritizing the children’s need to share their painful experiences (P1, P2). Dialogue, when children wish it, contributes to strengthening relationships between class members and promotes a positive mood (P8).

In this context, P1 stated that she seeks to bring students with similar experiences together so that they do not feel lonely because of the trauma they carry. As P6 pointed out, these are experiences that hinder the process of acquiring new knowledge. Furthermore, most of them emphasized the need for a systematic presence of a psychologist in the school, so that teachers and students could refer them to manage severe traumas and crises. The close cooperation of members of the school community who face similar problems was noted to have a catalytic role, as through their discussions, they jointly find ways to manage the challenges they face (P1, P3, P9). In particular:

P1: “… I tried to create a small community, to bring the children together. For these children who had difficult experiences… I tried to bring them together … and talk about their life stories and understand that they’re not alone… We also have a psychologist at school, and five people mentoring us over the last four years…”

P9: “Of course, most children don’t want to talk about their traumatic experiences. However, we discuss these experiences with Greek students, and how we can address them… we discuss these topics with our colleagues as well.”

According to the teachers, the absence of a permanent psychologist in every multicultural school who knows the students and their history constitutes a major problem. The participants noted that the fragmented presence of specialists and monitoring of students with intense psychological trauma make the work of teachers more difficult (P1, P8). As a result, insecurity and fear of the emergence and incorrect management of the mental trauma of migrant and refugee students prevail (P3, P6). Teachers also mentioned the lack of support and training in dealing with such situations. This situation burdens them psychologically and leaves them relatively unprotected (P6).

P6: “Yes, training of course. Because many times, some training from someone, either a psychologist, for example, I don’t know, or someone who has more experience…”

P8: “Of course, I should say that in general schools lack the issue of psychological support. I see a big gap there and not only for these children…. And there is no psychological support every day. There is a psychologist who comes back from the schools and comes once a week, so this person has a lot of children to see and doesn’t have the time to connect with the school community, with the daily life of the school and, with the children. So, it’s superficial.”

## 4. Discussion

The collected data confirm that despite the cultural and socio-economic differences, the teachers participating in the research attempt to promote inclusion and trauma management. By incorporating elements of the student’s cultural identity into the learning process, which they base on group activities and practices, they cultivate a climate of trust crucial for children’s psychosocial and cognitive development. Gay’s culturally sensitive teaching approach ([19]) is based on the principle that both teachers’ and students’ experiences should intrude into the learning context. When teachers appreciate students’ prior experiences and cultural heritage, students become more engaged in the learning experiences, demonstrate better academic success, and contribute to establishing cross-cultural communication and applying cultural congruity in class. Hence, [20] ([20]) encourages teachers to foster students’ critical processing of the status quo and to decode the hidden forces that perpetuate inequalities.

The analysis supports the theory of the multilingual teaching model as a means for teaching the target language, which, although a great challenge, becomes a crucial tool for acquiring new knowledge and integration. Teachers underlined the importance of psychological support for psychologically traumatized migrant and refugee students and pointed out the necessity of hiring permanent psychologists and having a broader supportive framework ([51]).

In addition, the participants seem to have realized from their own experience the importance of highlighting the cultural backgrounds of students to motivate them to participate in the learning process. This finding is consistent with Cummins’s research on the highlighting of pupils’ identity elements to promote their willingness to learn ([11], [8]). Through the promotion and sharing of aspects of their identity, their motivation to broaden their cognitive horizons and socialize is strengthened. At the same time, teachers mentioned that a genuine team spirit of confidence is built that is founded on respect for the diversity and uniqueness of students ([43]). Despite the innate tendency to integrate into groups of fellow nationals that create a sense of security, teachers pointed out that they seek to cultivate relationships of trust between class members regardless of their cultural backgrounds ([53]).

Teachers have also stated that promoting experiential learning through creative activities that highlight and at the same time mitigate the cultural differences of class members contributes to the development of a team and cooperative spirit. Projects, such as identity texts and various playful activities related to art, function as a motivation for participation in the team and as tools for intellectual and spiritual development. This fact echoes the aspect that through the involvement of their cultural backgrounds in the creation of their personal life book and through sharing with their classmates, the bonds between class members are strengthened, and all voices are empowered ([43]; [11]; [27]; [29]; [41]).

At the same time, participants emphasized the importance of creating a climate of trust between educators and students, as well as between children, so that they can share their personal stories. Implementing and maintaining a daily routine sets boundaries and creates a climate of safe interaction among class members ([52]; [53]). In this way, pupils seem to experience their participation in the learning process as a safe and creative experience that enhances self-respect as well as mutual respect.

Furthermore, the participants stated that they allow pupils to use their linguistic repertoire. Otherwise, communication would be significantly limited, and the promotion of new knowledge would be hindered ([54]). This is a student population with a migrant and refugee background that is often associated with relevant language skills for survival reasons. In addition to their mother tongue, these students know English or French to some extent, depending on their country of origin. This approach is in line with [9]’ ([9]) concept of Common Underlying Proficiency and language interdependence, which usually highlights the positive effects of language transfer on learning. This model advocates that there is a strong correlation between bilingual learners’ literacy skills in their mother tongue and second language; this situational reality allows learners to develop in both languages ([7]).

In the context of communication and understanding of instructions and new knowledge, educators allow “translanguaging.” This practice allows for direct student participation, a sense of acceptance, and the promotion of multicultural identity, resulting in removing the obstacles that diversity often poses ([18]; [47]; [55]). Students become creative and critically thinking speakers who emphasize primarily the meaning and secondarily the form of linguistic communication. The stress of social, cultural, and linguistic adaptation seems to be reduced, resulting in increased student autonomy and self-confidence ([35]). At the same time, using native-speaking students as interpreters strengthens cooperation between children despite the risk of data corruption due to incomplete transfer.

In agreement with the research findings of [22] ([22]), the effort of some participants to involve the parents or guardians of the students in the learning process is based on the realization that it can be a catalyst for the psycho-spiritual development of the latter. An open school to diversity does not reject but integrates differences and is enriched by otherness, arousing the trust of students and guardians. Thus, the willingness to participate in it and spiritual development increases ([4]).

Pupils with a migrant and refugee background carry psychological trauma due to abusive experiences they experienced both in their country of origin, during their migration journey, and in their respective host countries. Teachers may come into contact with their traumatic past during the learning process. The participants in the research, therefore, emphasized the importance of building a climate of trust and safety within the school community, so that students begin to realize that trauma belongs to the past and take advantage of the opportunities of the present ([52]). With empathy and discretion for the sensitive personal data of the students, they allow traumatic experiences to emerge, if they wish to share them. As they are not experts, they avoid going into details that they may not be able to manage appropriately ([49]).

However, reaching out to psychologists is difficult as, due to the underfunding of Greek public education, their number is minimal in relation to the needs of these students ([40]). For this reason, they primarily address colleagues who face similar challenges and share their experiences. It seems, therefore, that the power of a well-functioning school community functions as a counterbalance to systemic inadequacies and that teachers, by exchanging suggestions and ideas through their experiences, find new ways to deal with multiple challenges ([40]).

Students with post-traumatic stress disorder (PTSD) usually do not talk about their traumatic experiences, which hinders their psycho-spiritual development. Pre- or post-migration psychological trauma has been found to inhibit their cognitive and psychosocial development ([22]). This fact further complicates the work of teachers who, as non-experts, are unable to penetrate the barrier of terror that these vulnerable students may have experienced and have caused them to become alexithymic and disconnected from the present and new, potentially healing experiences ([52]). They seem to feel psychologically vulnerable and inadequate, as they are called upon to manage, without the systematic support of psychologists, issues that are not related to the specialty of an educator ([13]; [17]; [44]). In this regard, research findings reveal that most teachers in Germany feel unprepared to meet the needs of refugee children, whereas 20% of teachers in the Netherlands with over 18 years of experience in mainstream schools reported significant challenges in managing students’ trauma. Additionally, studies in several European and North American countries have shown that although many programs acknowledge the importance of trauma care, there is a widespread lack of adequate training for teachers ([51]).

In this vein, the shortcomings of the Greek education system—limited number of reception classes and special needs classes, limited and temporary teaching staff, outdated training, and monocultural teaching model—constitute notable obstacles to the fulfillment of the role of teachers ([38]; [31]; [1]; [30]). More creative and inclusive pedagogical methods and practices are needed that will utilize students’ pre-existing linguistic repertoire ([21]). Combined with the voluminous and strictly oriented exam material for the cognitive development of students, the educational task becomes even more difficult ([2]; [41]).

Finally, [5] ([5]) present a picture of the situation prevailing in multicultural education in Greece over the last decade, focusing on the obstacles and offering suggestions for their removal. Inadequate education and training of teachers, the lack of appropriate materials and infrastructure, and the large number of students of different cultural backgrounds constitute the main challenges for teachers. The above picture is complemented by the findings of [56] ([56]), who adds the lack of adequate coordination between Non-Governmental Organizations and the Greek state regarding the management of the huge refugee flows, situations that are exacerbated by the economic crisis that the country has gone through. [30] ([30]) point out that in the modern digital era, global capabilities are developing, while intercultural competencies are weakening. Despite the fact that teachers attempt to utilize the cultural background of their students through interactive teaching methods, they recognize the shortage of adequate training.

## 5. Conclusions

This study aimed to explore the strategies employed by the participants to address the diversity and psychological trauma of migrant and refugee students. Consistent with related research, participants highlighted the significance of cultural backgrounds and how these can be integrated into the learning process. To foster an environment conducive to communication and learning, teachers incorporate inclusive activities that reflect students’ cultural elements. Multimodal media and creative projects, such as identity texts, offer a safe space for students to express their previously unheard voices. These innovative practices are seen as effective in promoting both academic progress and personal growth.

While some participants initially embraced a dominant monocultural and monolingual teaching approach, their experiences revealed the benefits of a more inclusive language teaching method. By valuing students’ linguistic diversity, they found that students acquired the target language more easily, as opposed to prohibiting the use of their native languages. In fact, by showing interest in learning words from their students’ languages, some teachers shifted power dynamics and empowered students, boosting their enthusiasm and self-confidence, which are often lacking in these marginalized student groups.

According to interviewees, managing the psychological trauma of vulnerable students is a significant challenge. In the absence of permanent psychologists and social workers, teachers are left to handle students’ post-traumatic stress and psychological instability on their own. Some expressed feeling abandoned by the State and burdened by the emotional weight of their students’ traumatic experiences. However, they emphasized the importance of teachers supporting one another and sharing their experiences and challenges. They also recognized the value of setting boundaries for students’ mental well-being. Establishing a daily routine and structured activities provides a sense of security for students, whose lives are otherwise dominated by uncertainty.

When addressing the mental trauma that migrant and refugee students may face, teachers stressed the need for permanent mental health professionals to serve as a stable point of reference for both students and staff. They believe that creating a climate of trust and acceptance has a healing effect on psychological trauma. In doing so, they feel they would be able to more effectively help students overcome the trauma that hinders their cognitive and psychosocial development.

Finally, the study acknowledges its limitations, including the small sample size of participants and the researchers’ personal beliefs in favor of diversity and equality in education, which may have influenced the positive responses from participants. To gain a more comprehensive understanding of multicultural education in Greece, further qualitative and quantitative research, as well as increased awareness among the relevant authorities, is necessary.

## Data Availability

The submission of the thesis has been accepted and filed at DSpace and has the following identification: https://apothesis.eap.gr/archive/item/220347?lang=el (accessed on 25 March 2025).

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
