# Peer review of "Building Resilience: The Critical Role of Multicultural Education to Cope with the Psychological Trauma of Migrant and Refugee Youth"

_behavsci, 2025, doi:10.3390/bs15070916_

Round 1
Reviewer 1 Report
Comments and Suggestions for Authors
Dear authors
this is a very important topic, you have chosen for this paper.
Anyway, I do have to point to some points that need revision: First of all, there are some typos and some section headings (3.1. subsection) that should be deleted (I guess).
The paper needs some more information on the Greek school system and on the age of the pupils, to help understanding your critique on the “dominant monocultural and monolingual” doctrine – maybe referring to a law on integration or on schooling. Once you also mention the pressing timeframe set by the ministry, what is this referring to? It would also help to understand why you say, the curriculum is very strict.
Furthermore, a definition of the terms multicultural, multicultural class and multicultural education should be included.
In the introduction you mention “that education is crucial for facilitating the inclusion of refugees …. “ this whole paragraph sounds correct, some sources would surely support that feeling.
I do have problems understanding the reasoning about the connection of team spirit and cultural exchange, the example quotes you provide point maybe more to the failing or the challenges for team spirit then the success.
Similar holds for the section on team spirit through shared activities.
And the last section on team spirit 3.1.3 even points to cliques and security groups.
I read your interpretation as too positive. I would even challenge the underlying theme that cultural identity can promote team spirit – I have the feeling there is a step missing (maybe the class contract that you mentioned is the most successful intervention for a team spirit). But it is also not clear what team spirit should refer to, is it an identity building beyond cultural differences in a diverse class, or is it a self-positioning in smaller groups inside or beyond classes. I am sure you do have a clear understanding what you mean with team spirit.
In the section “discussion” you point out that participants propose the restructuring of multicultural education – also with a focus on training. This last part I did not see in the quotes or mention before.
Please also provide a source for the information on the impact of underfunding of the education system, psychologist are missing (it could also be a task of the health provision system…)
All in all I think you could cut down a bit on the length of the quotes and put more focus on reasoning of your interpretation. It can be that deeper knowledge of the Greek education system enables the reader to follow your interpretation, or that other/more quotes are needed to make your interpretation clear.
Comments on the Quality of English LanguageThere are some typos and some sentences are difficult to understand.
Sometimes, I guess, alternative words would be more fitting - like siblings instead of brothers in section 3.1.4
I just needs a thourough review.
Author Response
Dear reviewer,
First of all, we would like to thank you for the important feedback we have been provided with in order to further improve the quality of our text. We have edited the text based on your comments. All changes within the text are highlighted in bold yellow.
Regarding your comment on typos and some section headings, we would like to inform you that they have been deleted. In addition, some important information on Greek school system and Greek Legislation Framework and on its restricted implementation has been added as well. Regarding your third comment, a definition of the terms multicultural, multicultural class and multicultural education have been included, whereas the content of the paragraph “that education is crucial for facilitating the inclusion of refugees …. “has been justified by relative in-text citations.
Also, it was deemed necessary based on your comment on the research data of the first axis to focus on how inclusion is pursued in the multicultural classroom and not so much on team spirit. We would also like to mention that s reference to teachers training is made in lines 376-377, and a relevant extract from the interviews has been added (lines 379-380). In addition, a source for the information on the impact of underfunding of the education system has been added. Some quotes have been cut down and some other have been added to make our interpretation clear.
Finally, an attempt was made to improve the English version of the text by replacing some words/phrases with more suitable ones.

Reviewer 2 Report
Comments and Suggestions for Authors
Please, find my review attached.
Regards,

Author Response
Dear reviewer,
First of all, we would like to thank you for the important feedback we have been provided with in order to further improve the quality of our text. We have edited the text based on your comments. All changes within the text are highlighted in bold yellow.
We would like to inform you that the research purpose has been added to the introduction part so that it is distinct from the beginning of the reading of the text. In addition, and based on the comments of the 2nd reviewer, information about the Greek educational context has been added, which necessitates the elaboration of this research study.
Furthermore in the introduction and discussion part, the literature analysis on cultural responsive teaching, cultural competence, cultural congruity, cultural deficit thinking, and Common Underlying Proficiency has been strengthened. Also, information and in-text citations on the implementation of multicultural education in Greece and the monolingual and monocultural dimension of education have been added through the review of relevant research at the national level.
In addition, in the description of the survey sample, some information has been added to outline the profile of the participants and the target students. The reference to the concept of in-depth education has been worded to avoid misinterpretation.
Finally, an attempt was made to improve the English version of the text by replacing some words/phrases with more suitable ones.

Round 2
Reviewer 1 Report
Comments and Suggestions for Authors
Thank you for considering the previous review.
Author Response
We would like to thank you for your valuable conttibution!
Kind regards,
the authors
Reviewer 2 Report
Comments and Suggestions for Authors
I attach a file for my, by now minor comments. Some corrections and reasoning is still needed.

Author Response
Dear reviewer,
we would like to thank you again for your valuable comments and for your contribution to further improve our text.
The suggested minor changes are highlighted in green.
Kind regards,
The authors